# Risk Evaluation of Chemical Clogging of Irrigation Emitters via Geostatistics and Multivariate Analysis in the Northern Region of Minas Gerais, Brazil

Gustavo Lopes Muniz [1], Agda Loureiro Gonçalves Oliveira [1], Maria Geralda Benedito [2], Nicolás Duarte Cano [1], Antonio Pires de Camargo [1,*] and Ariovaldo José da Silva [1]

[1] Faculdade de Engenharia Agrícola, Universidade Estadual de Campinas, Campinas 13083-970, SP, Brazil
[2] Companhia de Desenvolvimento dos Vales do São Francisco e do Parnaíba (CODEVASF), Montes Claros 39400-292, MG, Brazil
* Correspondence: apcpires@unicamp.br

**Abstract:** In this study, we analyzed the hydrogeochemistry of 350 underground wells in the northern region of the state of Minas Gerais, Brazil, for water-chemical parameters that may contribute to the chemical clogging of emitters in drip irrigation systems. Risk class maps were generated for each parameter, and the area was classified based on the water characteristics, considering the degree of water-use restriction in micro-irrigation (i.e., no, moderate, and severe restriction). Inverse distance-weighted, random forest, and ordinary kriging methods were used as interpolation methods. Moreover, a multivariate analysis was conducted to analyze the results. Pearson's correlation coefficient showed a strong and significant correlation between pH and carbonates, hardness, total dissolved solids (TDS), and electrical conductivity (EC) and between TDS and EC. Principal component analysis revealed that most of the variations in the water quality of the wells could be explained by water–rock interactions with the consequent dissolution of minerals. The principal components were natural sources of ionic salt groups, dissolution of minerals rich in alkaline cations, chemical weathering of iron–magnesium minerals, and increased pH with the conversion of bicarbonates into carbonates. In the parameter cluster analysis, three possible mechanisms that contribute to emitter clogging in the study area were identified: precipitation of calcium and magnesium salts; oxidation of iron and manganese ions forming oxides and insoluble hydroxides; an increase in pH, which converts bicarbonates into carbonates. Clustering analysis revealed the wells that are susceptible to clogging with the exact cause.

**Keywords:** hard water; hydrogeochemistry; micro-irrigation; principal components; clustering



## 1. Introduction

The northern region of the state of Minas Gerais in Brazil is characterized by low precipitation and high evapotranspiration owing to the scarce rainfall concentrated at certain times of the year and high temperatures typical of semiarid regions [1]. The spatial rainfall variability and irregular pattern of rain distribution during the year promote the adoption of irrigation practices to ensure crop production and socioeconomic development [2].

The northern region of Minas Gerais has the largest irrigated agriculture project in Latin America, Project Jaíba, covering a total irrigable area of 107,600 ha, thereby demanding a high amount of water to meet the crop water requirements [3]. Surface water has been the major source of local agriculture for decades. However, with increased scarcity of surface water and pressures from management agencies in the region, groundwater has been increasingly used for human, industrial, and agricultural needs and has become important for maintaining agricultural–industrial sustainability [4]. In addition to the limited availability of surface water, the ease of access to underground water reserves owing to the improvements in drilling techniques has led to an increase in groundwater

usage for various purposes in the region, thereby significantly increasing its demand in recent years.

Drip irrigation is considered one of the appropriate irrigation techniques because it can provide high water-use efficiency in irrigation [5–8]. However, most emitters used in drip irrigation (i.e., drippers) have narrow flow sections (approximately 1 mm$^2$), which may favor clog formation [9]. Partial or complete clogging of drippers has been considered as the major limiting factor for the life span, as well as distribution efficiency, of a micro-irrigation system [10]. Dripper clogging is one of the major problems in the drip irrigation system [11–14], and many farmers in the region have abandoned this system early because of increased operation and maintenance costs and low service life.

Clogs form in the emitters owing to physical, chemical, and biological agents, and determining the cause of clogging is complex because of the interactions of these agents in water [15–18]. Water filtration and chemical treatment are used in drip irrigation systems to prevent emitter clogging [19–22]. Filtration systems are effective in removing suspended solids larger than the size of the filter element or medium. However, fine particles and dissolved chemicals pass through the filtration system, and under specific physicochemical conditions, they form precipitates and fouling in the components of the irrigation system, thereby impairing its operation and performance.

In the northern region of the state of Minas Gerais, groundwater is predominantly alkaline, with a tendency to form chemical precipitates because of the high levels of calcium, carbonates, and bicarbonates [23,24]. High pH values together with high atmospheric temperatures favor calcium carbonate precipitation; calcium carbonate is a low-soluble salt, which severely affects irrigation systems [15,25,26]. Thus, knowledge of water quality is fundamental for formulating preventive measures against dripper clogging.

Geochemical characteristics play an important role in groundwater quality and are significantly influenced by aquifer characteristics and anthropogenic activities [27–29]. Information on water quality is fundamental for proper analysis and management of irrigation systems, thereby extending their service life. The suitability of these systems in terms of water quality would avoid the frequent replacement of irrigation equipment, thereby increasing their sustainability. Knowledge of sites with low-quality water would result in stringent preventive measures such as better filtration systems, selection of emitters less susceptible to clogging, adoption of acidification routines, more frequent chlorination, and flushing of laterals.

Thus, in this study, we aimed to (1) characterize the groundwater of the northern region of Minas Gerais on the basis of the major chemical parameters that can cause emitter clogging; (2) generate maps using geostatistics to classify the areas according to the restrictions in water use based on criteria reported in the literature; (3) investigate the correlation among the analyzed chemical parameters and identify the major components and mechanisms that contribute to clog formation in emitters; (4) identify similar characteristics between the chemical parameters of the water and correlate them to possible clogging mechanisms; and (5) identify wells that are susceptible to clogging by clarifying the exact causes. This is the first study conducted in the region that addresses the chemical characteristics of water from a micro-irrigation perspective. The clogging risk maps generated by characterizing the chemical agents in irrigation systems can be used for the management of water resources for sustainable use and serve as a model to be applied in other places.

## 2. Materials and Methods

### 2.1. Study Area

The northern mesoregion of Minas Gerais (Figure 1) is a part of the 12 mesoregions of the state and comprises 89 municipalities, with a total area of 128,454,108 km$^2$ and population of 1,712,194 inhabitants. This mesoregion is considered the starting point of the semiarid region of the Brazilian Northeast. It is located in an area called a drought polygon, characterized by low productivity owing to its outdated technical base and climatic problems such as scarcity of rainfall [4].

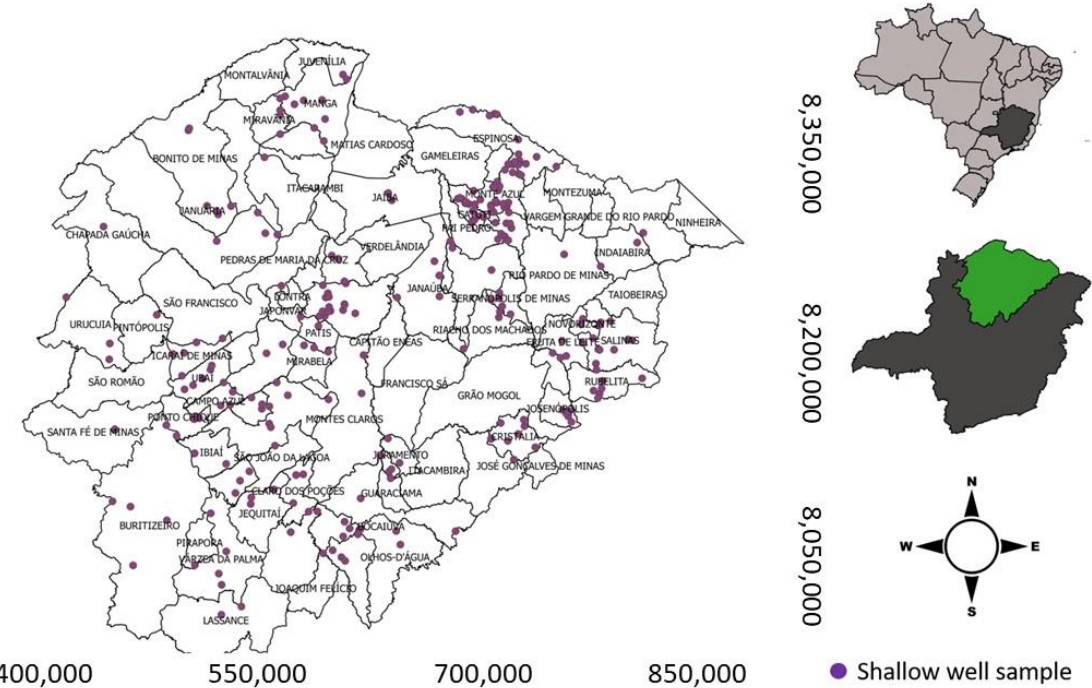

**Figure 1.** Map showing the study area and monitored underground wells.

According to the climatic classification of Köppen, 92% of the region has Aw climate (tropical savanna), which has a characteristic rainy season in summer (from November to April) and a dry season in winter (from May to October). The average temperature during the coldest month is above 18 °C. The average accumulated annual rainfall is less than 800 mm, and the Thornthwaite aridity index is less than 0.50. The daily percentage of water deficit in the region is higher than 60%, considering every day of the year [30].

The lithology of the study area primarily includes the rocks of the Urucuia Formation, Mata da Corda, Areado, and Bambuí Group. Approximately 69.7% of this region exhibits a flat to gently wavy topography. Geologically, the region has outcrops of Archean gneisses belonging to the crystalline basement, carbonate and terrigenous sedimentary rocks that make up the Bambuí Group of Neoproterozoic age, Urucuia Group sandstones of Cretaceous, and recent caprocks [1].

The region covers three sub-basins of the São Francisco River Basin: the Verde Grande River sub-basin (UPGRH SF10), the Jequitaí and Pacuí rivers sub-basin (UPGRH SF6), and the Pandeiros River sub-basin (UPGRH SF9). In this region, the Bambuí Aquifer, which is a karst, karst-fractured, and fractured-type aquifer, predominates with the prevalence of karst and karst-fractured characteristics [31]. Figure 2 shows the hydrogeological map of this region.

### 2.2. Water Quality Data

In this study, 350 wells in 61 municipalities distributed throughout the study area were selected to analyze the chemical characteristics of the groundwater. The waters of the wells that are in operation and monitored by the Geological Service of Brazil in partnership with the Institute of Management of Mining Waters (IGAM) were analyzed. All wells are intended for local supply of water. The analytical results obtained refer to a sampling period of four consecutive years. Sample collection, processing, and storage were performed according to standard procedures described in the standard methods for examining water and wastewater [32].

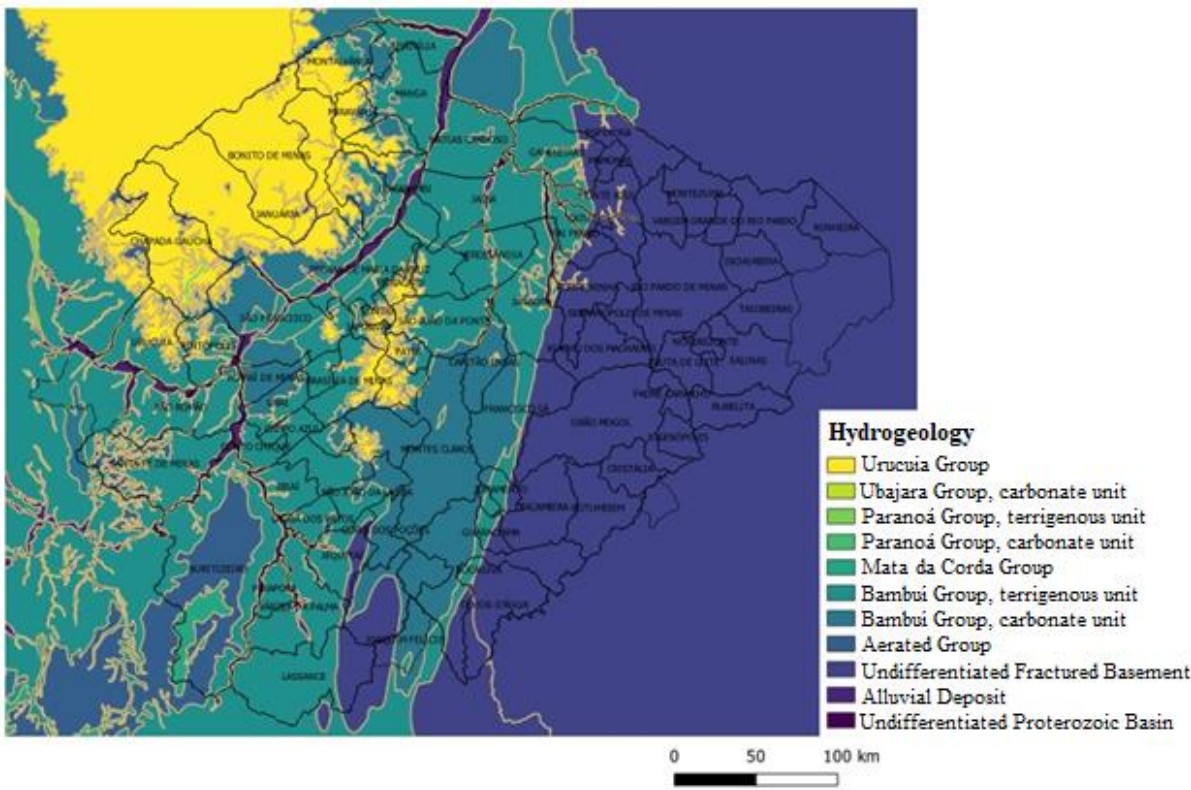

**Figure 2.** Map of hydrogeological domains of the study area.

The chemical parameters of the analyzed water indicated the risk of emitter clogging by chemical agents. The selection of these parameters was based on previous knowledge of water quality in the region and problems reported by residents and farmers and in the literature [23,33]. The total hardness, bicarbonates, carbonates, pH, iron, manganese, total dissolved solids (TDS), and electrical conductivity (EC) were analyzed. The pH and EC were determined via electrometric and conductive methods. The total hardness and carbonate and bicarbonate contents were determined via the titrimetric method. The iron and manganese concentrations were determined via atomic absorption spectrometry. TDS was determined via the gravimetric method. Trace metals were analyzed via atomic absorption spectrometry-direct air-acetylene flame method (APHA 3111 B).

Compounds such as hydrogen sulfide, phosphates, sulfates, silicates, and hydroxides, which can also form precipitates, were not considered in this study because their concentrations were extremely low and undetectable in the waters of most wells. In addition, only a few scientific reports have stated emitter clogging problems due to these compounds when well water is used for irrigation. Deep artesian wells may contain approximately 1 mg L$^{-1}$ hydrogen sulfide; however, this is uncommon; thus, the monitoring of total iron, TDS, and pH and estimation of the calcium carbonate precipitation tendency are performed in deep wells [34–37].

### 2.3. Area Classification Limits

Correlating the water quality results obtained in the laboratory with precipitation risks in the field is not an easy task. Information regarding the risk of precipitate formation for various combinations of cations and anions may be useful in determining whether clogging may occur [38]. For restricting the usage of groundwater for irrigation purposes on the basis of the quality indicator parameters, only the risk of emitter clogging by chemical agents was considered [26,36,38–40]. Thus, the areas were classified into three classes of water-use restriction: no restriction, moderate restriction, and severe restriction [41,42], and their respective limits are presented in Table 1.

**Table 1.** Classification scale of the degree of restriction of water use for micro-irrigation.

| Parameters | Degree of Restriction | | | Reference |
| --- | --- | --- | --- | --- |
| | None | Moderate | Severe | |
| pH | <7.00 | 7.00–8.00 | >8 | [38] |
| Total hardness (mg L$^{-1}$ as CaCO$_3$) | <150 | 150–300 | >300 | [38] |
| Iron (mg L$^{-1}$) | <0.2 | 0.2–1.5 | >1.5 | [38] |
| Manganese (mg L$^{-1}$) | <0.2 | 0.2–1.5 | >1.5 | [38] |
| Bicarbonate (mg L$^{-1}$) | <91.5 | 91.4–518.5 | >518.5 | [43] |
| Carbonate (mg L$^{-1}$) | <6.0 | 6.0–12.0 | >12.0 | [41] |
| Total dissolved solids (mg L$^{-1}$) | <500.0 | 500.0–2000.0 | >2000.0 | [38] |
| Electrical conductivity (dS m$^{-1}$) | <0.7 | 0.7–3.0 | >0.3 | [43] |

*2.4. Geostatistics*

Three interpolation methods were used to generate maps representing the spatial behavior of groundwater quality variables: inverse distance weighting (IDW), ordinary kriging (Ok), and random forest spatial (RFsp).

IDW is a deterministic method because it requires fewer calculations to meet specific statistical assumptions than other stochastic methods, such as kriging [44]. It can treat extreme values (outliers) in datasets compared to other spatial interpolation methods and is widely used for mapping areas using water quality data [42,45–47]. The weighted sum of the values of *N* known points gives the value of an unknown point. Equations (1) and (2) describe the formulas used in the IDW:

$$\widehat{Q_w} = \sum_{i=1}^{N} w_i Q_i \tag{1}$$

$$w_i = \frac{d_i^{-\alpha}}{\sum_{i=1}^{N} d_i^{-\alpha}} \tag{2}$$

where $\widehat{Q_w}$ is the unknown value of the variable, $Q_i$ is the known value of the sampled stations, $N$ is the total number of stations sampled, $w_i$ is the weight of each sampled station, $d_i$ is the distance of each station sampled to unknown points, and $\alpha$ is a parameter that controls the significance of known points in the interpolated values based on their distance, which is a positive real number [42]. The analysis followed a three-step procedure: (1) selection of water quality parameters and reclassification of pixel-based maps (raster graphic), (2) weight allocation for each water quality parameter, and (3) combination of weighted parameters and generation of final maps [47].

Kriging is a geostatistical estimator method that considers the spatial characteristics of autocorrelation of regionalized variables. It is used to interpolate the surface of a set of known scattered points, in which a continuous surface of values can be predicted between known points [48]. The regionalized variables require a certain spatial continuity, which allows the data obtained by sampling certain points to be used to parameterize the estimation of points where the variable value is unknown. However, if the variable lacks spatial continuity in the study area, estimating/interpolating using kriging has no logical sense [49]. Mathematical models provide information regarding spatial variation structure and kriging input parameters [48,50,51].

RFsp is a statistical method of supervised machine learning based on building a set of decision trees or regression trees. This method represents the knowledge embedded in an existing dataset in the tree format. Tree nodes are created based on the characteristics (features) of the dataset. The number of trees (*n* tree) and explanatory variables (*m* try) are two important parameters in the RFsp training process [52].

The results obtained using these three interpolation methods were compared using the cross-validation criteria. This technique is widely employed in problems where the aim of modeling is prediction. The observed and estimated values were used to evaluate the

effectiveness of the interpolation method. The root mean square error (*RMSE*) was used as the standard for comparison in this study (Equation (3)):

$$RMSE = \sqrt{\sum_{i=1}^{N} \frac{\left(\widehat{Y}_i - Y_i\right)^2}{N}} \quad (3)$$

where $\widehat{Y}_i$ and $Y_i$ are the estimated and observed values at location $i$, respectively, and $N$ is the number of observations. For an appropriate estimator, the *RMSE* should be as small as possible [53,54].

### 2.5. Multivariate Analysis

Multivariate data analysis methods are widely used to reduce the number of variables with minimal loss of information and are widely used in hydrogeochemical studies [27,28,42,46]. In this study, data were analyzed via Pearson's correlation analysis (at a statistical significance level of 5%), principal component analysis (PCA), and cluster analysis.

Pearson's correlation analysis was performed to understand the relationship between different water quality parameters. In this study, a correlation coefficient (r) $\geq$ 0.7 was considered strong, r between 0.4 and 0.7 was considered moderately strong, and r < 0.4 was considered weak correlation [42,55]. Normality and homoscedasticity tests were performed, and the need to transform the variables was verified. The Shapiro–Wilk normality test was performed at 5% statistical significance. All variables presented a normal distribution; thus, data transformation was unnecessary.

PCA is a technique for pattern recognition that attempts to explain the variance of a large set of intercorrelated variables and transforms them into a small set of independent variables (main components) with minimal loss of information [46,56,57]. In PCA, the Kaiser criterion was applied [58]; only factors with eigenvalues greater than or equal to 1 were accepted as possible sources of variance in the data [46]. PCA was performed using eight parameters (chemical variables) analyzed in the 350 wells studied. In PCA, absolute load values > 0.75, between 0.75 and 0.50, and between 0.50 and 0.30 were classified as "strong", "moderate", and "weak", respectively [57,59]. The PCA results were compared and confirmed via Pearson's correlation analysis [46,57]. The components considered in the PCA represented the total number of major factors contributing to the risk of emitter clogging in the northern region of Minas Gerais.

Cluster analysis is a statistical technique used to classify elements, such that similar elements remain within the same group and distinct elements remain in different groups [60]. A function of distance or similarity is used to define the similarity or difference between the elements. The analysis can be divided into two types: hierarchical clustering (HC) and non-hierarchical clustering. These two cluster analysis methods were used in this study.

In HC, we sought to group the chemical parameters of water to identify similar characteristics between them and relate these characteristics to possible clogging mechanisms. Ward's method and Euclidean distance were used for similarity measurements, which are widely used in hydrochemical studies [27,28,46].

In non-hierarchical clustering, we attempted to group the sampled wells ($N = 350$) according to the different chemical parameters that indicate emitter clogging risk. Thus, it was possible to identify which parameters highly affected the formed groups. This type of analysis is essential because, when the characteristics of the formed groups have been identified, specific preventive measures can be suggested regarding the risk of chemical clogging of emitters for a given group, considering the parameters that most influenced its formation. The k-means algorithm was applied in this technique, and Euclidean distance similarity coefficients were used. K-means is a non-hierarchical clustering heuristic that seeks to iteratively minimize the distance of the elements to a set of k centers given by $\chi = \{x_1, x_2, \ldots x_k\}$. The distance between a pi point and cluster set, given by $d(p_i, \chi)$,

is defined as the distance from the point to the center closest to it. The function to be minimized is given by Equation (4) [61]:

$$d(P, \chi) = \frac{1}{n} \sum_{i=1}^{n} nd(p_i, \chi)^2 \qquad (4)$$

The algorithm depends on a user-defined parameter (*k* = the number of groups). In this study, we used *k* = 4, which is defined according to the characteristics of the studied parameters.

Multivariate statistics were performed using the Minitab software (version 19). The values of parameters were transformed to a mean of zero and unit variance. This was to avoid overestimating or underestimating the weight of a parameter owing to the differences in the measurement scale [62].

The presentation assigned a number corresponding to the municipality and the sample point within it to each well. Thus, each site was renamed with a code. For example, point M1P1 refers to Well 1 in Municipality 1. A summary of the methodology is presented in the form of a flowchart in Figure 3.

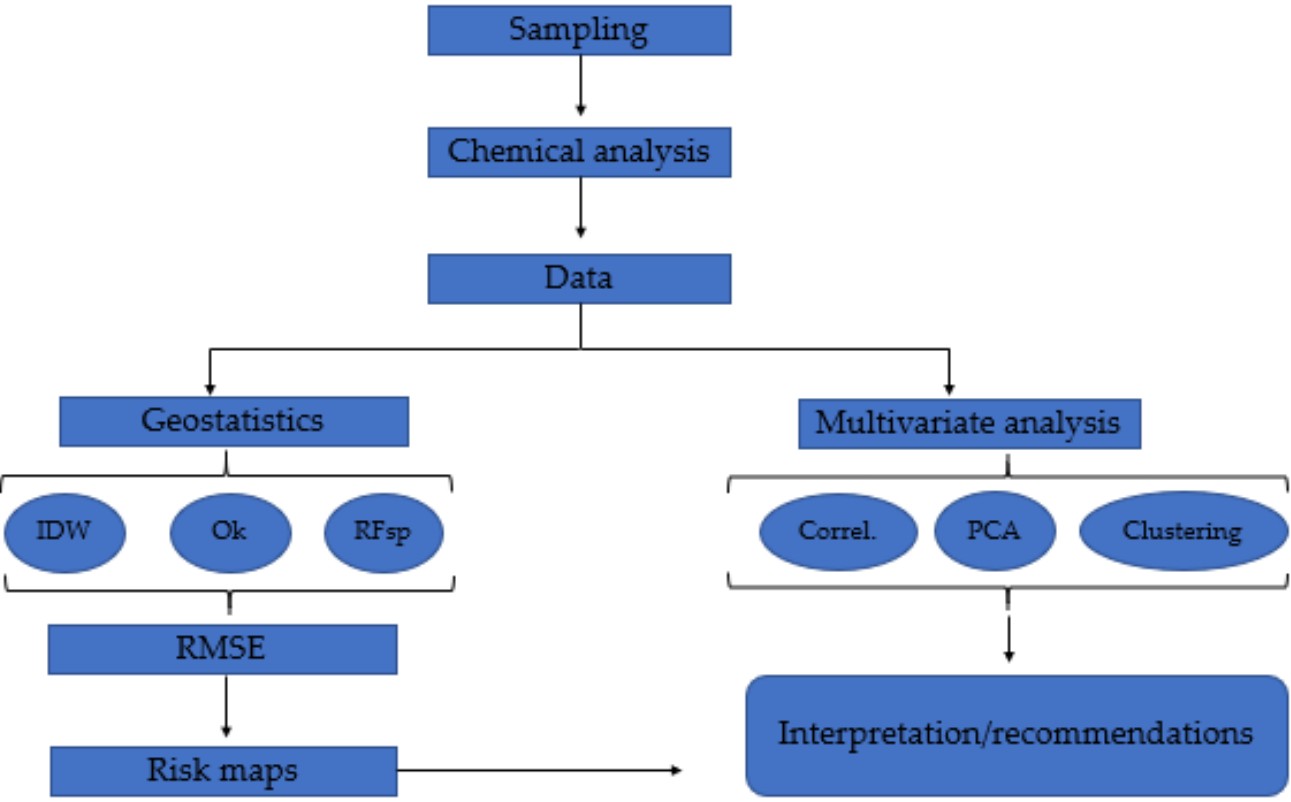

**Figure 3.** Study methodology flowchart.

## 3. Results and Discussion

### 3.1. Characterization of Samples

Table 2 presents the average and dispersion measures for each parameter. Significant variations were observed in these values, indicating the different types of groundwater in the region, the quality of which can be influenced by various factors. The spatial distribution of wells in different geological formations (Figure 2) produces a wide range of values for the same parameter [63].

**Table 2.** Descriptive statistical analysis of water samples from the monitored wells.

| Parameters | Max | Min | $\bar{x}$ [1] | $S_x$ [2] |
|---|---|---|---|---|
| pH | 8.90 | 4.52 | 7.76 | 0.52 |
| Total hardness (mg L$^{-1}$ as CaCO$_3$) | 2316.00 | 13.88 | 292.30 | 304.30 |
| Bicarbonate (mg L$^{-1}$) | 917.00 | 11.0 | 216.0 | 114.0 |
| Carbonate (mg L$^{-1}$) | 11.00 | <0.01 | 2.00 | 2.00 |
| Iron (mg L$^{-1}$) | 22.60 | 0.04 | 0.49 | 1.67 |
| Manganese (mg L$^{-1}$) | 3.13 | 0.02 | 0.23 | 0.36 |
| Total dissolved solids (mg L$^{-1}$) | 5018.00 | 26.00 | 541.40 | 622.00 |
| Electrical conductivity (dS m$^{-1}$) | 6.96 | 0.05 | 0.78 | 0.86 |

Note: [1] $\bar{x}$ = average; [2] $S_x$ = sample standard deviation.

Most sampled wells generally had water with a moderate risk of clogging irrigation emitters, as indicated by the analyzed parameters (Table 1). Total hardness and iron values were high, with maximum of 2316.0 mg L$^{-1}$ (as CaCO$_3$) and 22.6 mg L$^{-1}$, respectively. The study area is known to have waters with high hardness, and many farmers have reported problems of emitter clogging primarily due to CaCO$_3$ precipitates, which is a salt of low solubility that precipitates in waters with high hardness, alkalinity, and pH. These results indicate that the water from these wells requires attention for its use in micro-irrigation systems because it poses a high risk of dripper clogging.

### 3.2. Characteristic Maps

Table 3 presents the *RMSE* results of the three interpolation methods. We used the interpolation method, whose metric appears in bold, to draw a map of the respective parameters.

**Table 3.** Root-mean-square error of the different interpolation methods.

| Interpolation Method | pH | Hardness (mg L$^{-1}$ as CaCO$_3$) | Bicarbonate (mg L$^{-1}$) | Carbonate (mg L$^{-1}$) | Iron (mg L$^{-1}$) | Manganese (mg L$^{-1}$) | TDS (mg L$^{-1}$) | EC (dS m$^{-1}$) |
|---|---|---|---|---|---|---|---|---|
| RFsp | 0.52 | **164.38** | **90.58** | 2.24 | **0.50** | 0.46 | 551.20 | 0.73 |
| IDW | **0.51** | 174.84 | 99.99 | 2.19 | 0.80 | 0.35 | 492.77 | 0.69 |
| Ok | 0.52 | 211.78 | 107.88 | **2.14** | 0.60 | **0.35** | **390.60** | **0.68** |

Note: RF = random forest; IDW = inverse distance weighting; Ok = ordinary kriging. Bold indicates the interpolation method used.

Ref. [64] observed that RFsp, IDW, and Ok presented similar performances in interpolating evapotranspiration data prediction maps. According to them, regarding computational performance, an interpolation model should be created in short time. The IDW also achieved the lowest prediction times. IDW and Ok are two widely known spatial interpolation algorithms owing to their low estimation errors [64]. According to Machado et al. [65], RFsp is sensitive to different training dataset selection strategies despite being considered a robust spatial predictor model.

Figure 4 shows the interpolated maps for each parameter. The green regions indicate the areas in which there is no restriction for using water in drip irrigation systems, considering the risk of emitter clogging by chemical agents; blue regions indicate that, in these areas, the risk degree is moderate; and, finally, areas in red indicate severe water-use restriction, and waters in these areas are the most susceptible to emitter clogging by chemical agents.

#### 3.2.1. Total Hardness

Approximately 33% of the wells had water with hardness values above 300 mg L$^{-1}$ as CaCO$_3$, while 36% of wells had water with values between 150 and 300 mg L$^{-1}$ and 31% of the wells had water with values below 150 mg L$^{-1}$. Hard water contains relatively high concentrations of calcium and magnesium salts [66]. Approximately 30% of the study area had wells with water having hardness values higher than 300 mg L$^{-1}$ as CaCO$_3$. In these locations, using water for micro-irrigation is severely restricted because this water

is susceptible to the precipitation of calcium and magnesium salts. Areas with moderate water-use restrictions were observed primarily in the central region. Areas with no water-use restrictions were observed in the western region and a small strip covering the eastern region. The hardness values are a function of the presence of calcareous and dolomitic rocks, which occur in a large part of the studied region (Figure 2). Because the water is in direct contact with these rocks, groundwater is typically saturated with these ions, which increases the chances of their precipitation [66]. Relatively high concentrations of calcium and magnesium in water can precipitate in the form of low-solubility salts, particularly carbonates. The formation of calcium and magnesium carbonates is favored under the conditions of high temperature and pH [67,68]. Calcium and magnesium carbonates can precipitate in filters, pipes, and emitters when the water has pH values higher than 7.2 and a high temperature and hardness [24,36]. When calcareous water is used in drip irrigation, emitter clogging is frequent because of water evaporation around the drippers, forming a crust and thereby favoring clogging [39,69].

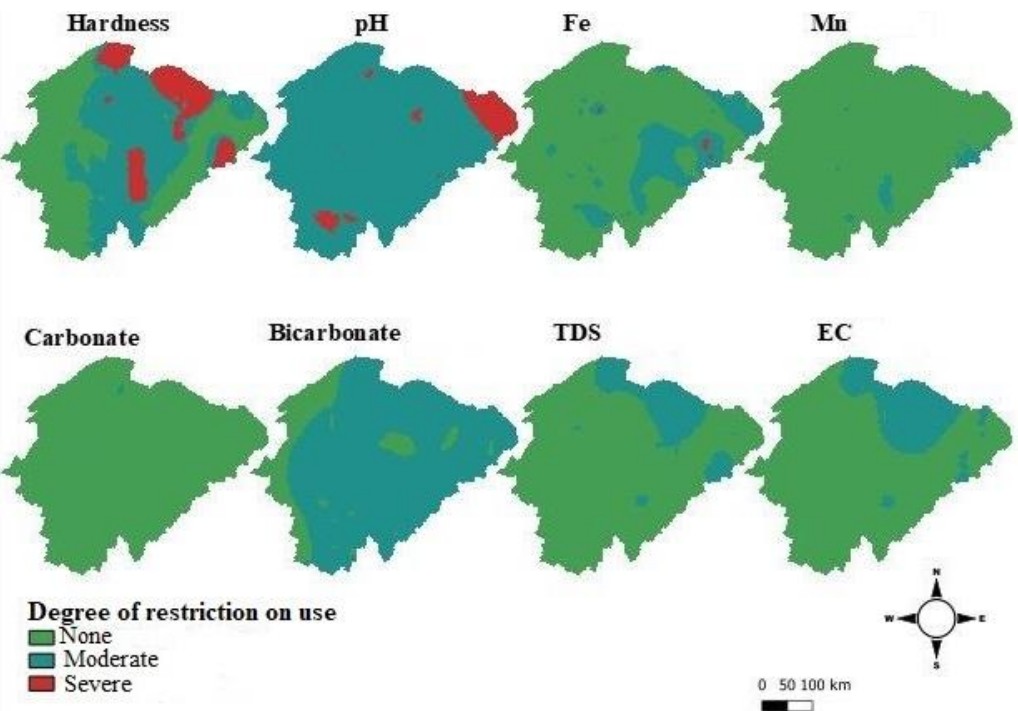

**Figure 4.** Degree of restriction on the use of groundwater for irrigation based on hardness, pH, iron, manganese, carbonate, bicarbonate, TDS, and EC.

The total hardness parameter alone may not necessarily indicate the precipitation risk of calcium or magnesium carbonates. The precipitate formation depends on the carbonate and bicarbonate concentrations of the medium. In addition, it requires high pH values for the reaction to occur. The concentration of dissolved solids and the water temperature also directly influence the formation of these salts. The risk of calcium carbonate precipitation can be evaluated using saturation indices, which encompass all these parameters and predict the risk of precipitation based on the concept of saturation. The most commonly used index is the Langelier saturation index (LSI) [70]. However, the LSI only indicates the risk of calcium carbonate precipitation, disregarding magnesium carbonate. Thus, LSI was not considered because the total hardness of water corresponded to the sum of calcium and magnesium ion concentrations in the water.

3.2.2. pH

As presented in Table 2, the pH of the water ranges from 4.52 to 8.90, with a mean value of 7.76, which is closely related to alkaline mineral sources in the aquifer system [27].



Of the wells sampled, 31% had water with pH values above 8.00; 63% of wells had water pH values between 7.00 and 8.00, and only 6% of the wells had water with pH values below 7.00. Regarding emitter clogging hazards in drip irrigation systems, waters with pH < 7.00 have no use restrictions, waters with pH values between 7.00 and 8.00 present moderate restrictions, and pH values above 8.00 have severe restrictions of use [38]. According to Figure 4, most of the study area offers a moderate risk of emitter clogging, considering the pH values. In the far east, the degree of water-use restriction was severe.

The pH value of groundwater depends on the carbon dioxide–carbonate–bicarbonate balance [71]. Thus, water with high pH values and high concentrations of bicarbonate and carbonate requires careful attention if calcium and magnesium ions are present because of the potential risk of precipitation of calcium and magnesium carbonates, which are the salts with low solubility and potential clogging risk of drippers [15]. Similarly, iron, manganese oxides, and hydroxides form at a pH greater than 7.00. Therefore, pH control in drip irrigation systems is indispensable to maintaining their potential performance [38,72].

### 3.2.3. Bicarbonates and Carbonates

Regarding carbonate concentrations in water, the entire area was classified to have no water-use restriction as most wells, approximately 93% of the sampled wells, had waters with carbonate concentrations lower than 6.0 mg $L^{-1}$. Carbonate ions were present in waters with pH values higher than 8.30, and bicarbonate ions were dominant in waters with pH values lower than 8.30. Regarding bicarbonates, approximately 85% of the area had waters with a moderate degree of restriction for using in drip irrigation systems, except for the extreme west, which had waters with bicarbonate concentrations lower than 91.5 mg $L^{-1}$, which corresponds to approximately 12% of the total wells sampled. Bicarbonate and carbonate ions are potential causes of drip clogging because they react with cations, such as calcium and magnesium, thereby forming precipitates and fouling in emitters and piping. Carbonates and bicarbonates originate in water because carbonate and silicate minerals are dissolved by carbonic acid [73], and the concentration of these ions in water is entirely dependent on the pH and $CO_2$ partial pressure in the system [15]. The levels of bicarbonates and carbonates are the major chemical factors contributing to the alkalinity of water [74]. High alkalinity is problematic for drip irrigation systems because of water hardness and calcium and magnesium concentrations. Relatively high ion concentrations in water tend to precipitate Ca and Mg as carbonates. In the case of high alkalinity, hardness, and pH, ions form insoluble minerals that cause fouling, partially or totally reducing the discharge of drippers [15,19].

### 3.2.4. Iron and Manganese

Most of the studied areas presented no water-use restriction due to iron concentrations, except for the extreme east and part of the center-east, which presented a moderate degree of restriction. Approximately 63.4% of the wells had water with an iron content below 0.2 mg $L^{-1}$, while 30.5% of the wells had water with an iron content between 0.2 and 1.5 mg $L^{-1}$. Notably, the same part in the extreme east, with moderate restriction regarding iron, presented severe restrictions regarding pH. Therefore, this area would be the most likely to present problems of iron fouling in the components of the irrigation system. In these areas, preliminary water treatment is essential to remove iron, prolong the life of the irrigation system, and avoid future problems. Regarding Mn, most of the area had no water-use restriction, with 70% of the wells presenting water with manganese values below 0.2 mg $L^{-1}$, and 28% presenting water with values between 0.2 and 1.5 mg $L^{-1}$. However, even at low concentrations, Fe and Mn are oxidized in the presence of chemoautotrophic bacteria, creating problems in micro-irrigation systems [40,75–77]. Soluble and reduced forms of iron (ferrous ions or $Fe^{2+}$) are present in well water in many places. Sludge formation can occur at $Fe^{2+}$ concentrations above 0.4 mg $L^{-1}$ and is typically associated with $Fe^{2+}$ oxidation and $Fe^{3+}$ precipitation because it is insoluble. The entire oxidation process is performed by ferrous bacteria in water, such as Gallionella, Leptothrix, Toxothrix,

Crenothrix, and Sphaerotilus, and other non-filamentous aerobic bacteria of the genera Pseudomonas and Enterobacter [38]. Primary blocking agents are typically sticky bacterial sludge that adheres to suspended solids and not necessarily the precipitated iron itself. The presence of iron in water may also favor the growth of several other bacteria, such as filamentous Vitreoscilla, non-filamentous Pseudomonas, and Enterobacter, which can cause emitter clogging owing to their gross mass and, in this case, may constitute a clogging problem of biological origin.

Reactions of iron with organic complexing agents are also necessary for forming sludge. Specific iron–organic complexes can adhere to glass and plastic materials, such as scratches in the emitters and walls of micro-irrigation pipes, even when the bacteria are already dead owing to the action of chlorine. The reddish-brown spots formed in the rocks when phenolic compounds were dissolved in well water containing iron show the tenacious adherent properties of the complexed iron. This discoloration can be avoided by treating water with chlorine (NaOCl—0.5 mg $L^{-1}$) or hydrogen peroxide ($H_2O_2$—5.0 mg $L^{-1}$) [37,38,78]. This study did not evaluate the presence of iron-oxidizing bacteria and iron- and manganese-complexing agents.

### 3.2.5. Total Dissolved Solids (TDS) and Electrical Conductivity (EC)

Water–rock interactions are responsible for groundwater salinity in the study area [73,79]. In this study, groundwater TDS values ranged from 26.0 to 5018.0 mg $L^{-1}$, with a mean of 541.4 mg $L^{-1}$. The TDS showed greater dispersion, with a standard deviation of 621.28 mg $L^{-1}$, which supposedly results from several factors, such as aquifer heterogeneity and human disturbances in the area, which was also observed by Liu et al. [27]. Regarding the degree of restriction, most of the region presented no water-use restriction regarding TDS (approximately 67.8% of the studied wells), except for one portion in the northern region, which presented a moderate restriction of water use (28.5% of the wells).

The water EC ranged from 0.05 to 6.96 dS $cm^{-1}$, with a mean value of 0.78 dS $cm^{-1}$. Similarly, regarding EC, most of the area is classified as having no restriction, except for a far north portion. Approximately 64.5% of the wells had water EC values below 500 mg $L^{-1}$, approximately 32% of the wells had water with EC between 500 and 2000 mg $L^{-1}$, and only 3.5% of the wells had water with EC values above 2000 mg $L^{-1}$. The EC directly reflects the precipitation and dissolution processes that occur in the aquifer system, owing to the common-ion effect, because the solubility of a compound depends on the nature and concentration of other substances, primarily ions, in the mixture, which can directly influence the increase or decrease in EC [66].

### 3.3. Multivariate Analysis

#### 3.3.1. Correlation Analysis

Pearson's correlation coefficient was used to investigate the relationship between the chemical parameters of the wells (Table 4).

**Table 4.** Pearson's correlation matrix of chemical parameters.

| Parameter | pH | $HCO_3^-$ | $CO_3^{2-}$ | Hardness | Fe | Mn | TDS | EC |
|---|---|---|---|---|---|---|---|---|
| pH | 1.00 | | | | | | | |
| $HCO_3^-$ | 0.117 * | 1.00 | | | | | | |
| $CO_3^{2-}$ | **0.714 *** | 0.356 * | 1.00 | | | | | |
| Hardness | 0.098 | 0.497 * | 0.104 | 1.00 | | | | |
| Fe | −0.027 | −0.050 | −0.033 | −0.054 | 1.00 | | | |
| Mn | −0.046 | 0.031 | −0.043 | 0.167 * | 0.467 * | 1.00 | | |
| TDS | 0.091 | 0.407 * | 0.138 * | **0.896 *** | −0.050 | 0.179 * | 1.00 | |
| EC | 0.087 | 0.417 * | 0.129 * | **0.871 *** | −0.079 | 0.115 * | **0.972 *** | 1.00 |

Note: * Significant at a level of 0.05. The bold number indicates a strong correlation (r ≥ 0.7) at 0.05.

The correlation between variables can provide valuable information on the geochemical processes that occur in aquifer systems. Correlation coefficients (r) greater than 0.7 show a strong correlation, whereas correlation coefficients from 0.4 to 0.7 indicate a moderate correlation [27]. Table 4 presents a strong linear correlation of carbonate concentration with water pH (r = 0.714), hardness with TDS (r = 0.896) and EC (r = 0.871), and EC with TDS concentration (r = 0.972). All these correlations were positive, indicating that the increase in one parameter leads to an increase in the other, and they presented statistical significance at the 5% probability level.

The parameters that characterized saline water (TDS and EC) had the highest correlation with each other (r = 0.972). The dissolved salt content has a proportional relationship with the EC of the water, and the salt content can be estimated via EC measurement at a given temperature in water with pH values between 6.5 and 8.5 [32]. Previous studies on groundwater have also reported high correlations between TDS and EC parameters [42,80].

The presence of local limestone and basaltic rocks explains the strong correlation between hardness and EC because they are susceptible to dissolution processes in the presence of carbonic acid, thereby releasing ions that can increase both the hardness value and EC of water. Dissolution combines water with carbon dioxide ($CO_2$) from the atmosphere and enriches the soil. The result is a carbonic acid ($H_2CO_3$) solution or acidic water, responsible for epigenic karstification [81]. Waters with high EC values are associated with limestone and basaltic rocks, which suffer chemical weathering more easily than granites and quartzites. Basaltic and limestone rocks primarily consist of magnesium, iron silicates, and calcium carbonates, which potentially cause emitter clogging in drip irrigation systems. Therefore, areas with high EC values require more attention when selecting, designing, and maintaining irrigation systems [82]. We observed a strong positive correlation between EC and calcium and magnesium ions, which are the main contributors to water hardness. According to Varol [82], both cations are responsible for water mineralization.

A strong correlation between the carbonate content and pH was also observed. This correlation can be explained by carbonates present only in water with a pH greater than 8.3. Under these conditions, part of the inorganic carbon appears as carbonates. The bicarbonate parameter shows a moderate linear correlation with hardness, TDS, and EC. The correlation of bicarbonate with hardness is expected in waters with pH values lower than 8.3 as carbonate rocks, primarily calcareous, dissolve in these waters with dissolved $CO_2$-releasing calcium. This increases the hardness of the water and the bicarbonate concentration as the pH will favor the presence of this element owing to the calco-carbonic equilibrium conditions [66]. The other correlations were considered weak (r < 0.4) and insignificant.

### 3.3.2. Principal Component Analysis (PCA)

PCA was conducted to combine correlated variables into a small number of principal components. Each component indicated one of the major factors contributing to the clogging risk of irrigation emitters. Based on the Kaiser criterion [58], only the first three factors that explained 80% of the total variance were used in PCA (Figure 5).

Table 5 presents the matrix of the eigenvalues and eigenvectors of the components considered for each parameter. Bold values indicate component loads greater than 0.5; these values are considered significant contributors to the total variance [46].

The first component (PC1) was responsible for 40.2% of the total variance and presented strong positive loads (> |0.5|) for hardness, TDS, and EC. PC1 represented the natural sources of ionic groups of salts [82]. The second component (PC2) was responsible for 21.7% of the total variance and presented substantial positive loads for pH and carbonate parameters, representing the source of physicochemical variations and dissolution of minerals rich in alkaline cations [73,83]. Finally, the third component (PC3) presented substantial positive charges for iron and manganese parameters, indicating similar geochemical behaviors, which are controlled by the chemical weathering of iron–magnesium minerals

in Plio-Pleistocene aquatic sediments and by the dissolution of basaltic rocks [73,84]. PC3 represented elements susceptible to redox reactions and highly depended on the water pH, redox potential (Eh), temperature, and dissolved oxygen concentration.

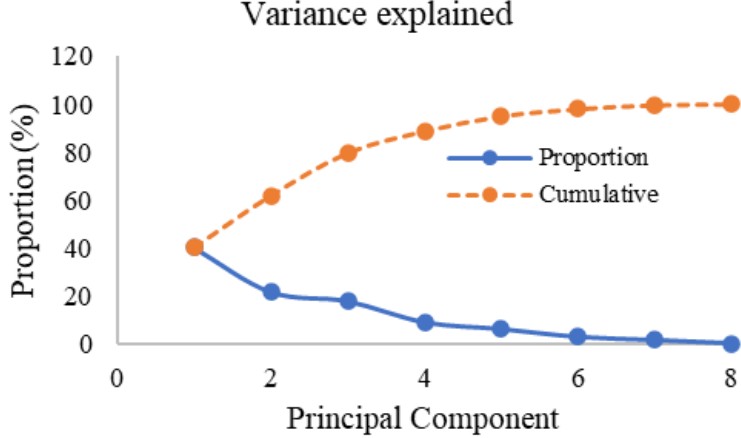

**Figure 5.** Percentage of cumulative and proportional variance explained by the components.

**Table 5.** Results of the principal component analysis.

| Parameter | PC1 | PC2 | PC3 |
|---|---|---|---|
| pH | 0.127 | **0.621** | 0.246 |
| $HCO_3^-$ | 0.349 | 0.148 | 0.002 |
| $CO_3^{2-}$ | 0.183 | **0.630** | 0.235 |
| Hardness | **0.516** | −0.157 | −0.069 |
| Fe | −0.034 | −0.214 | **0.687** |
| Mn | 0.100 | −0.296 | **0.628** |
| TDS | **0.528** | −0.142 | −0.053 |
| EC | **0.522** | −0.129 | −0.097 |
| Eigenvalue | 3.215 | 1.735 | 1.426 |
| Proportion | 0.402 | 0.217 | 0.178 |

Note: PC1 = principal component 1; PC2 = principal component 2; PC3 = principal component 3. Bold values indicate component loads greater than 0.5.

In general, the PCA results indicated that most variations in the water quality of the wells could be explained by the natural sources of ionic groups of salts, water–rock interactions, consequent dissolution of minerals rich in alkaline cations, and chemical weathering of iron–magnesium minerals.

Figure 6a shows a biplot of PC1 and PC2. This graph shows how variables represented by vectors impact the relationships between PC1 and PC2, which together explain 61.9% of the total variance. The wells plotted in the biplot chart significantly overlapped, making it difficult to identify them. In this case, we made generalized inferences to avoid the misinterpretation of the wells. Most of the wells were located near the origin (zero), thereby presenting intermediate values of the analyzed parameters. Some points sampled in certain municipalities diverged from others; in these locations, it can be assumed that there are anthropogenic sources that can cause strong interference in groundwater quality [42,46,56].

As presented in Table 5, the parameters that were strongly correlated with PC1 were hardness, TDS, and EC. The municipalities in the extreme right of the first and fourth quadrants in Figure 6a presented high values of these parameters, indicating that the water from the wells in these municipalities presents a high risk of emitter clogging, considering salinity and calcium and magnesium concentrations in the water.

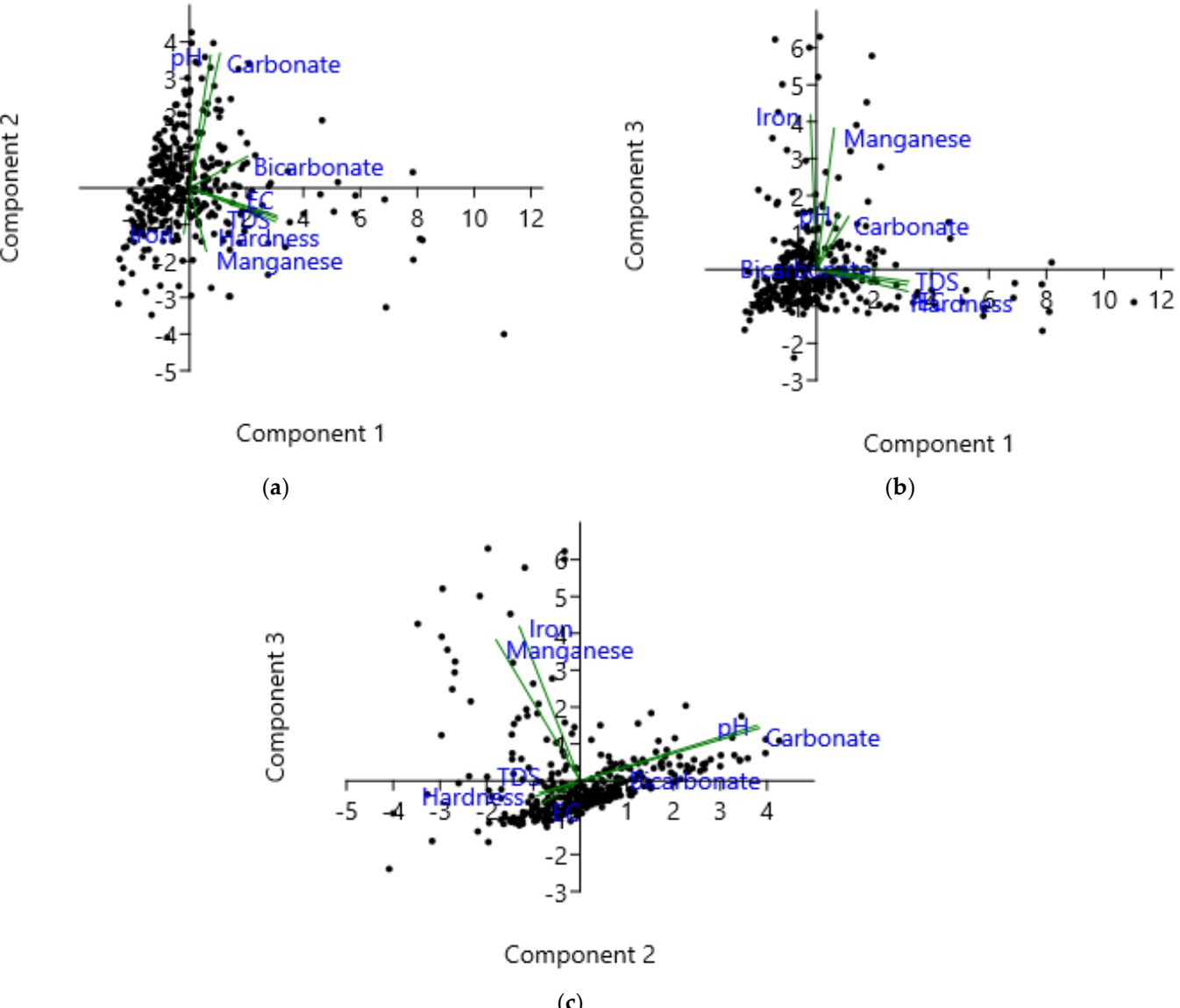

**Figure 6.** Bivariate plot of (**a**) PC1 versus PC2, (**b**) PC1 versus PC3, and (**c**) PC2 versus PC3, with observations and variables.

From the analysis of PC2, the wells in the first and second quadrants (Figure 6a) presented water with high pH values and carbonate concentrations, thereby presenting an alkaline character. Therefore, in these places, calcium, magnesium, and iron concentrations must be tested because alkaline pH favors the precipitation of these ions. Stringent clogging prevention practices should be adopted in these places because chemical precipitation phenomena are more common at pH values > 7.00 [38,66,78].

Figure 6b shows the biplot of PC1 versus PC3, which explains 58.03% of the total variance. PC3 was strongly correlated with the iron and manganese parameters; the wells that were plotted in the first and second quadrants had the highest values of iron and manganese in the water. The iron and manganese precipitation in water is highly pH-dependent and occurs in waters with alkaline pH; the PC2 versus PC3 plot (Figure 6c) shows relevant information about redox reactions. Figure 6c, regarding PC2, shows that the wells in the first and fourth quadrants have waters with alkaline characteristics; therefore, these wells are more susceptible to iron and manganese precipitation. Through the analysis of PC3, which strongly correlated with these ions, it was verified that wells containing water with high concentrations of iron and manganese are in the first and second quadrants.

Therefore, it can be concluded that the wells in the first quadrant are the most susceptible to iron and manganese precipitation because redox reactions occur at pH values greater than 7.00 in the presence of oxygen. Equations (5) and (6) show the oxidation reactions of iron and manganese by the action of oxygen, forming low-solubility precipitates.

$$4Fe(HCO_3)_2 + O_2 + 2H_2O \rightarrow 4FeOH_3 + 8CO_2 \tag{5}$$

$$2Mn(HCO_3)_2 + O_2 + 2H_2O \rightarrow 2Mn(OH)_4 + 4CO_2 \tag{6}$$

Figure 6c shows several wells in the fourth quadrant. Despite presenting waters with high pH values, these wells are less likely to suffer from redox reactions of iron and manganese ions because they show low concentrations of these elements. Similarly, despite the high concentrations of iron and manganese, the problems of dripper clogging by iron and manganese precipitates in the wells in the second quadrant are mitigated owing to low pH values.

The PCA results corresponded with Pearson's correlation analysis results (Table 3), which was also observed by Sudhakaran et al. [57].

### 3.3.3. Cluster Analysis

Multivariate cluster analysis has typically been used in hydrogeochemical studies [28,46,56,73,79,82,83,85]. In this study, cluster analysis was performed using the hierarchical method with Euclidean distance and Ward's method to group chemical parameters and identify similar characteristics between them that can relate them to possible mechanisms that contribute to emitter clogging. Furthermore, the non-hierarchical k-means clustering method was used to group the sampled wells (observations) with similar characteristics and to present waters with a risk of emitter clogging owing to a specific group characteristic, as suggested by Lima et al. [46].

Figure 7 shows the result of clustering the variables using the 50% dendrogram cutting criterion [46], resulting in three different clusters defined according to distance. It was verified that the parameters were grouped similar to the formation of the principal components, thereby reinforcing the existence of different mechanisms that can contribute to the clogging of irrigation emitters in the study area. Group 1 included the pH and carbonate parameters, reflecting the weathering process of alkaline rocks. Group 2 included iron and manganese, reflecting the chemical weathering process of iron–magnesium minerals and representing the elements susceptible to redox reactions. Group 3 comprised the parameters of hardness, bicarbonate, TDS, and EC, reflecting the natural sources of ionic groups of salts that confer salinity to the water.

On the basis of the results shown, it can be concluded that three main mechanisms contribute to the clogging of emitters in the northern region of Minas Gerais. These mechanisms may be related to the dissolution processes of the aquifer rocks. The mechanisms are as follows: (1) an increase in the pH and conversion of bicarbonate into carbonate can increase the supersaturation of the carbonate system; (2) increased concentrations of iron and manganese in water, which are susceptible to oxidation and formation of insoluble oxides and hydroxides; and (3) increased concentrations of calcium and magnesium ions, which precipitate with carbonate, forming salts with low solubility. Notably, an increase in pH favors the precipitation of calcium and magnesium salts and the precipitation of iron and manganese oxides and hydroxides [66], which are the major chemical precipitates found in the labyrinths of drippers.

Table 6 presents the results of the non-hierarchical clustering of wells using k-means. This method can identify wells, within the same group, with water that poses the risk of clogging emitters owing to the same chemical process. In the k-means method, the number of arbitrary groups (k) was equal to four.

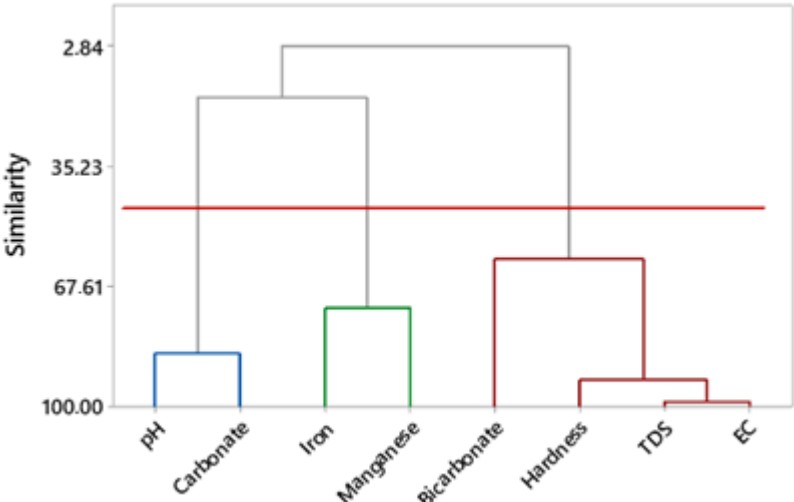

**Figure 7.** Dendrogram showing clustering of variables by Ward's method.

**Table 6.** Results obtained in clustering using k-means with k = 4.

| Group | Frequency | SQ | MD | MaxD | Distance between the Centroids of the Groups | | | |
|---|---|---|---|---|---|---|---|---|
| | | | | | 1 | 2 | 3 | 4 |
| 1 | 23 | 248.868 | 2.976 | 7.402 | - | 5.35 | 6.63 | 6.02 |
| 2 | 116 | 393.526 | 1.713 | 3.678 | 5.35 | - | 4.34 | 2.19 |
| 3 | 26 | 282.060 | 3.042 | 7.149 | 6.63 | 4.34 | - | 4.05 |
| 4 | 185 | 451.594 | 1.390 | 5.950 | 6.02 | 2.19 | 4.05 | - |

Note: SQ: sum of squares of the grouped; MD: centroid mean distance; MaxD: maximum centroid distance from.

As presented in Table 6, 23 wells belonged to group 1, 116 belonged to group 2, 26 belonged to group 3, and finally, the majority, 185 wells, belonged to group 4. The sum of squares within the cluster measures the variability of observations within each cluster. Clusters with smaller sum-of-square values exhibit less variability of observations within the cluster and are more compact than a cluster with a larger sum of squares. Similarly, clusters with the lowest mean centroid distance exhibit less variability of observations within the same cluster. The distance between the centroids of the clusters indicates how different the clusters are from each other, i.e., the greater the distance, the more significant the difference.

Figure 8 shows the distances between the centroids of the parameters, which represent the distances of the parameters to the centers of the groups formed, for the four groups.

On the basis of Figure 7, it can be established that Group 1 presents greater centroid distance values than the other groups for the indicator parameters of salinity and mineralization processes of limestone rocks: bicarbonate, hardness, TDS, and EC. This implies that the water from the wells belonging to this group presents a high risk of calcium and magnesium salt precipitation. This group presented a centroid distance value for the pH parameter near the origin, indicating that the pH of these waters is close to the average pH value of the sampled waters (7.76), characterizing the waters as alkaline, thereby exhibiting increased risk of precipitate formation. Group 2 contained wells with high pH and carbonate concentrations and a relatively high number of wells. Using these waters in drip irrigation systems requires stringent clogging prevention measures, such as applying acids to maintain a slightly acidic medium [38]. Group 3 contained wells that presented the most significant distance from the centroid to the iron and manganese parameters; therefore, when these waters are used for drip irrigation systems, some treatments should be performed to remove these ions. Finally, Group 4 contained wells that presented values closer to the centroid, and this group consisted of most of the sampled wells.

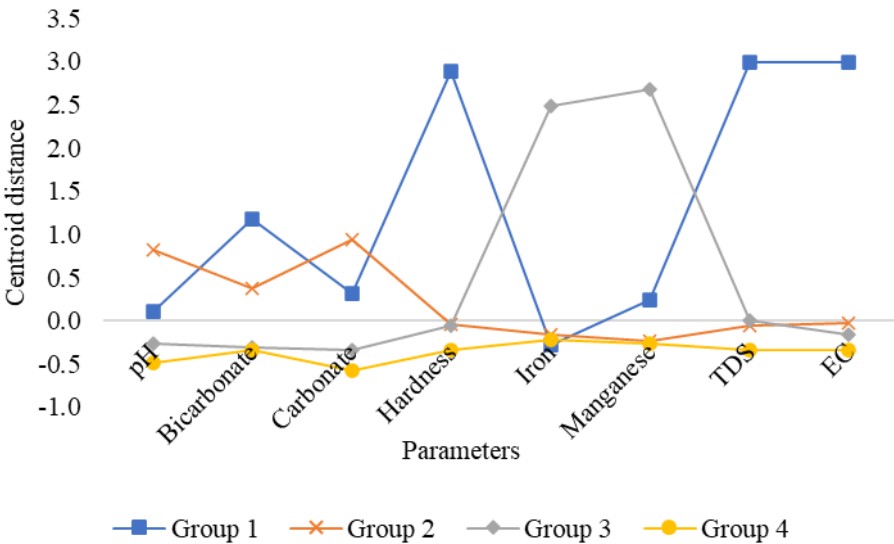

**Figure 8.** Distance from the centroid of the different variables for the formed groups.

## 4. Conclusions

Water samples from 350 underground wells located in the northern region of Minas Gerais were analyzed and characterized on the basis of the parameters that pose risks of emitter clogging by chemical agents. The study area was classified to three according to the degree of water-use restriction: severe, moderate, or no restriction. Considering the risk of chemical precipitation, most of the studied area may present problems in drip irrigation systems and consequently emitter clogging.

Pearson's correlation coefficient showed a strong correlation between pH and carbonate concentration, total hardness, TDS, and EC, and between TDS and EC. The major factors that contribute to the clogging risk of emitters by chemical agents were identified via PCA. All these factors are related to the water–rock interactions and the consequent dissolution of minerals from the rocks that constitute the aquifer: natural sources of ionic groups of salts, chemical weathering of iron–magnesium minerals, and dissolution of minerals rich in alkaline cations that contribute to the increase in pH and conversion of bicarbonate into carbonate.

In the HC method, parameter groups with similar characteristics related to the possible mechanisms of clogging were identified: Group 1, formed by the pH and carbonate parameters; Group 2, formed by the iron and manganese parameters; and Group 3, formed by the hardness, bicarbonate, TDS, and EC parameters. Wells within the same group that are susceptible to the exact cause of clogging were identified in the cluster analysis via k-means clustering.

From the centroid distances of the different parameters for the groups formed, it was revealed that the waters of the studied wells, when used for drip irrigation systems, must go through treatment stages. Moreover, prevention and maintenance measures should be strictly followed in the management of these systems. These results provide useful information and/or offer some recommendations about the prevention of clogging for farms, particularly for users and those who design drip irrigation systems.

**Author Contributions:** Conceptualization, G.L.M., A.L.G.O. and A.P.d.C.; methodology, G.L.M. and A.L.G.O.; software, G.L.M. and A.L.G.O.; validation, G.L.M. and A.L.G.O.; formal analysis, G.L.M., A.L.G.O., N.D.C., M.G.B., A.P.d.C. and A.J.d.S.; investigation, G.L.M.; resources, G.L.M. and A.P.d.C.; data curation, G.L.M. and A.L.G.O.; writing—original draft preparation, G.L.M. and A.P.d.C.; writing—review and editing, G.L.M., A.L.G.O., N.D.C., M.G.B., A.P.d.C. and A.J.d.S.; visualization, G.L.M. and A.P.d.C.; supervision, A.P.d.C.; funding acquisition, A.P.d.C. All authors have read and agreed to the published version of the manuscript.

**Funding:** This research was funded by the Coordenação de Aperfeiçoamento de Pessoal de Nível Superior-Brasil (CAPES) [grant number 88882.434695/2019-01].

**Data Availability Statement:** Not applicable.

**Conflicts of Interest:** The authors declare no conflict of interest.

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
