# Peer review of "Risk Evaluation of Chemical Clogging of Irrigation Emitters via Geostatistics and Multivariate Analysis in the Northern Region of Minas Gerais, Brazil"

_water, doi:10.3390/w15040790_

Round 1

Reviewer 1 Report

All corrections, comments and recommendations  are shown on the text using Word Tracking System. All can be seen there.

Author Response

The authors thank you for reviewing the manuscript. All suggestions were accepted. Professional service for English proofreading was hired to improve the manuscript.

Results and Discussion section was improved according to your recommendations.

A sentence was added to the end of Conclusions section, as suggested.  

Reviewer 2 Report

The subject of the work is very topical.

Author Response

The authors thank you for reviewing the manuscript. Professional service for English proofreading was hired to improve the manuscript.

Reviewer 3 Report

1.      The manuscript needs professional English editing. There are a lot of grammar errors.

2.      Line 579-581: sentence confusing, paraphrase.

3.      Line 35: remove “and”

4.      Line 92: add colon after “to”

5.      Line 102: remove “one of the” wrong grammar

6.      Line 138: correct to small letters “Standard Method”

7.      In the materials and methods section, author should indicate how the trace metals were analyzed

8.      Line 437: replace “CO2withCO2

Author Response

The authors thank you for reviewing the manuscript. Professional service for English proofreading was hired to improve the manuscript.

The method used to trace metals was added to the Methodology, as requested.

Reviewer 4 Report

Comments:

The author(s) put considerable effort to write the paper. It must be encouraged. I tried to mention some of my comments/suggestions below. In my opinion, this paper is acceptable for publication with major revision.

·         The work seems to be good but more novelty behind the proposed approach needs to be added at the end of the introduction, in order to improve the significant contributions of the authors.

·         Improve the quality of figures 1 and 2, kindly.

·         The general structure of the methodology should be presented in the form of a flowchart that shows the tools, methods, and the reason for their use in achieving the study goals.

·         In Figure 7, what is meant by center distance is the average distance of the parameter of that group to the center? It must be clearly stated in the article.

·         In the results section, it is necessary to present and interpret a general map of the risk zoning of water consumption for agricultural purposes in terms of its origin and usage limits.

The results section needs a fundamental revision in terms of analyzing the obtained results from the used methods and their integration fo

Author Response

The authors thank you for reviewing the manuscript.

Introduction was improved.

Figures 1 and 2 were attached as individual files to ensure better quality of these images.

Sentences related to Figure 7 were rephrased.

Results and Discussion section were improved, as suggested.